

# Comprehensive analysis of the microbiome in *Apis cerana* honey highlights honey as a potential source for the isolation of beneficial bacterial strains

Pham T. Lanh[1,2], Bui T.T. Duong[1], Ha T. Thu[1], Nguyen T. Hoa[1] and Dong Van Quyen[1,2]

[1] Laboratory of Molecular Microbiology, Institute of Biotechnology, Vietnam Academy of Science and Technology, Hanoi, Vietnam
[2] University of Science and Technology of Hanoi, Vietnam Academy of Science and Technology, Hanoi, Vietnam

## ABSTRACT

**Background**. Honey is a nutritious food made by bees from nectar and sweet deposits of flowering plants and has been used for centuries as a natural remedy for wound healing and other bacterial infections due to its antibacterial properties. Honey contains a diverse community of bacteria, especially probiotic bacteria, that greatly affect the health of bees and their consumers. Therefore, understanding the microorganisms in honey can help to ensure the quality of honey and lead to the identification of potential probiotic bacteria.

**Methods**. Herein, the bacteria community in honey produced by *Apis cerana* was investigated by applying the next-generation sequencing (NGS) method for the V3–V4 hypervariable regions of the bacterial 16S rRNA gene. In addition, lactic acid bacteria (LAB) in the honey sample were also isolated and screened for *in vitro* antimicrobial activity.

**Results**. The results showed that the microbiota of *A. cerana* honey consisted of two major bacterial phyla, *Firmicutes* (50%; *Clostridia*, 48.2%) and *Proteobacteria* (49%; *Gammaproteobacteria*, 47.7%). Among the 67 identified bacterial genera, the three most predominant genera were beneficial obligate anaerobic bacteria, *Lachnospiraceae* (48.14%), followed by *Gilliamella* (26.80%), and *Enterobacter* (10.16%). Remarkably, among the identified LAB, *Lactobacillus kunkeei* was found to be the most abundant species. Interestingly, the isolated *L. kunkeei* strains exhibited antimicrobial activity against some pathogenic bacteria in honeybees, including *Klebsiella spp.*, *Escherichia coli*, *Enterococcus faecalis*, *Pseudomonas aeruginosa* and *Staphylococcus aureus*. This underscores the potential candidacy of *L. kunkeei* for developing probiotics for medical use. Taken together, our results provided new insights into the microbiota community in the *A. cerana* honey in Hanoi, Vietnam, highlighting evidence that honey can be an unexplored source for isolating bacterial strains with potential probiotic applications in honeybees and humans.

Corresponding author
Dong Van Quyen,
dvquyen@gmail.com

**OPEN ACCESS**

## INTRODUCTION

Honey is a sweet and highly viscous substance produced by bees, and an important food and therapeutic agent for human beings since ancient times (*Israili, 2014*; *Kačániová et al., 2009*; *Khalili Samani et al., 2021*; *Stavropoulou et al., 2022*) due to its antimicrobial, antioxidant, and anti-inflammatory properties and ability to improve the immune system (*Ewnetu, Lemma & Birhane, 2013*; *Lee, Churey & Worobo, 2008*; *Schell et al., 2022*). Honey for human consumption is mainly produced by *Apis mellifera*, *A. andreniformis*, *A. caucasica*, *A. cerana*, *A. dorsata*, *A. florea*, and *A. indica* (*Israili, 2014*). The composition, physicochemical properties, and flavor of honey are closely associated with the floral source used by the honeybees, the regional and climatic environment, and beekeeping practice (*Bouhlali et al., 2019*; *Israili, 2014*; *Warui et al., 2019*).

Generally, honey contains about 600 compounds, including carbohydrates (about 95–97% of the dry weight of honey), proteins, lipids, and minerals, as well as trace elements, vitamins, choline, acetylcholine, free amino acids, $\beta$-carotene, lycopene, hormones, enzymes, antioxidants, organic acids, phenolic acids, and phenolic acid derivatives, acetophenone, benzaldehyde, $\rho$-cymene derivatives, pheromones, *etc*. (*Ewnetu, Lemma & Birhane, 2013*; *Israili, 2014*; *Stavropoulou et al., 2022*; *Warui et al., 2019*), and notably, bacterial strains producing a variety of metabolites with antimicrobial activity against the invasion of several pathogens (*Israili, 2014*; *Khalili Samani et al., 2021*; *Lee, Churey & Worobo, 2008*; *Schell et al., 2022*; *Stavropoulou et al., 2022*). The bacterial load in honey samples is varied and generally lower than 5 log CFU/g (the safety limits for use) (*Devi et al., 2021*) due to its physicochemical properties such as high sugar concentration (85%–95%), hyperosmolarity (15.0%–17.3%), low acidity (pH ranged from 3.2 to 4.5), hydrogen peroxide content and antibiotic activities (*Devi et al., 2021*; *Israili, 2014*; *Samarghandian, Farkhondeh & Samini, 2017*; *Silva et al., 2017*; *Stavropoulou et al., 2022*).

Bacteria in honey can be derived from primary sources, including the environment (air, soil, *etc.*), the honeybee diet (pollen, nectar, honey, *etc.*), the bee gut flora, or secondary sources of post-harvest contamination (*Kačániová et al., 2009*; *Khalili Samani et al., 2021*). Among these sources, pollen and nectar are the most important because most bees ingest them as their primary food, and they are commonly colonized by several bacteria and yeast (*Pozo et al., 2021*). Once collected by bees (honeybees, stingless bees, or bumblebees), nectar is initially stored and concentrated into honey, in which osmotolerant microorganisms may have a role in honey maturation during long-term honey storage. On the other hand, pollen is stored inside the nests or brood cells, then moistened by mixing it with secretions (containing honey, enzymes, and microbes) from the bee honey stomach and undergo different phases of microbial modification (*Lee, Churey & Worobo, 2008*; *Pozo et al., 2021*).

Common bacteria found in honey were *Bacillus* (*B. subtilis*, *B. licheniformis*, *B. badius*, *B. circulans*, *B. coagulans*, *B. pumilis*), *Enterococcus* (*E. faecium*, *E. hirae*), *Lactobacillus* (*L. plantarum*, *L. pentosus*, *L. kunkeei*), *Klebsiella*, and spores of *Clostridia, etc.* (*Khalili Samani et al., 2021*; *Saha, Ahammad & Barmon, 2018*). While some of these microorganisms may cause spoilage or fermentation of the honey, others may have beneficial effects on the health of honeybees (*Schell et al., 2022*; *Silva et al., 2017*). As honey is a versatile and

valuable natural substance that can be used for a variety of purposes, including as a food, a natural sweetener, a wound dressing, cough and sore throat remedy, and a skincare ingredient; therefore microbiological criteria in honey have been concerned with quality and safety (*Israili, 2014*; *Snowdon & Cliver, 1996*).

Previous studies have unveiled microbial profiles in *A. cerana* honey, yet a focused exploration of the distinctive microbial communities in *A. cerana* honey in Vietnam is lacking. The present study aims to explore the microbial community in the honey produced by *A. cerana* honeybees, a domesticated honeybee with a long history of beekeeping in Vietnam, by using the next-generation sequencing (NGS) for the V3–V4 regions of the 16S rRNA gene. Subsequently, attention is given to the isolation of lactic acid bacteria exhibiting the capability to combat some pathogenic bacteria in honeybees based on the insights garnered from the NGS results. The obtained results may provide insights into developing biological products to improve honeybees' health and novel strategies to ensure the quality of honey for better uses of honey in daily and medicinal purposes.

## MATERIALS & METHODS

### Honey sample

Ten honey samples from ten healthy *A. cerana* colonies were collected in Hanoi, Vietnam (20.04 N 105.5560 E) by the Research Center for Tropical Bees and Beekeeping, Vietnam National University of Agriculture, Hanoi, Vietnam. The colonies of *A. cerana* originated from their respective hives. The bees were provided with fresh frames to build comb and store honey. The wax and honey are sourced from the same colonies. The honeybee colonies with no symptoms were collected and tested to be free of *Nosema spp.* (*Chaimanee, Warrit & Chantawannakul, 2010*) and seven important honeybee viruses, including Sacbrood virus (SBV) (*Grabensteiner et al., 2001*), acute bee paralysis virus (ABPV) (*Benjeddou et al., 2001*), deformed wing virus (DWV) (*Yue & Genersch, 2005*), black queen cell virus (BQCV) (*Benjeddou et al., 2001*), Kashmir bee virus (KBV) (*Stoltz et al., 1995*), cloudy wing virus (CWV) (*Thu et al., 2016*), Israeli acute paralysis virus (IAPV) (*Thu et al., 2016*) by PCR or RT-PCR as described in a previous study (*Thu et al., 2016*). The experiments were carried out in duplicate. All pairs of primers are in Table S1. All the honey samples were collected on the same day and tested immediately after collection. Subsequent to the completion of the PCR analysis, the samples were subjected to assays for micro analysis.

### Total DNA extraction

A total of 30 g of each honey sample was dissolved in 15 ml of sterile water. The solution was incubated at 65 °C for 30 min, followed by centrifugation for 10 min at 5,000 rpm. The supernatant was discarded, and the pellets were utilized to extract total DNA using GeneJET Genomic DNA Purification Kit (Thermo Fisher Scientific, Waltham, MA, USA) according to the manufacturer's instructions. The DNA concentration and purity were determined by a Nanodrop 2000 UV-vis spectrophotometer (Thermo Fisher Scientific, Waltham, MA, USA), and DNA quality was checked by 1% agarose gel electrophoresis and visualized under UV light. The qualified DNA samples (containing 70 ng of each sample) were pooled for sequencing.

## PCR amplification and Illumina sequencing

The PCR amplification and sequencing were done by Chunlab, Inc. (Seoul, Korea). Briefly, PCR amplification was performed using primers targeting the V3–V4 hypervariable regions of the 16S rRNA gene with extracted DNA, using primers 341F (5′-TCGTCGGCAGCGTCAGATGTGTATAAGAGACAGCCTACGGGNGG CWGCAG-3′; and 805R (5′-GTCTCGTGGGCTCGG-AGATGTGTATAAGAGACAGGACT ACHVGGGTATCTAATCC-3′). Then, secondary amplification was carried out using a unique dual index with i5 forward primer (5′-AATGATACGGCGACCACCGA GATCTACAC-XXXXXXXXTCGTCGGCAGCGTC-3′; and i7 reverse primer (5′-CAAGCAGAAGACGGC ATACGAGATXXXXXXXX-GTCTCGTGGGCTCGG-3′) (X indicates the barcode region) to attach the Illumina NexTera barcode. The PCR product was confirmed using 1% agarose gel electrophoresis, visualized under a Gel Doc system (BioRad, Hercules, CA, USA), then purified and removed non-target products with the CleanPCR (CleanNA). The quality and product size were determined on a Bioanalyzer 2100 (Agilent, Palo Alto, CA, USA) using a DNA 7500 chip. Mixed amplicons were pooled, and the sequencing was performed at Chunlab, Inc. (Seoul, Korea), with the Illumina MiSeq Sequencing system (Illumina, San Diego, CA, USA) according to the manufacturer's instructions.

## 16S rRNA sequence processing

Sequence and microbiome analyses were performed with EzBioCloud, a commercially available ChunLab bioinformatics cloud platform (*Chun et al., 2007*; *Kim et al., 2012*; *Yoon et al., 2017*). The raw data was initially checked for quality (QC) and filtered out the low-quality reads (<Q25) using Trimmomatic v0.32 (*Bolger, Lohse & Usadel, 2014*). The quality-controlled reads were then merged to paired-end reads using PANDAseq (*Masella et al., 2012*); the primer trimmers were carried out by ChunLab's in-house program at a similarity cut-off of 0.8. The 16S rRNA non-specific amplicons were detected by HMMER's hmmsearch program (*Eddy, 2011*). The sequence-denoising process was performed by DUDE-Seq (*Lee et al., 2017*). The non-redundant reads were found by using UCLUST-clustering (*Edgar, 2010*). Sequences were then applied to taxonomic assignment by USEARCH using the EzBioCloud database version PKSSU4.0 and more precise pairwise alignment (*Myers & Miller, 1988*). The chimeras were detected by UCHIME (*Edgar et al., 2011*) and the non-chimeric 16S rRNA database from EzBioCloud. Species-level determined using the cut-off of 97% similarity of 16S rRNA gene sequences. The sequences were clustered by CD-HIT (*Fu et al., 2012*) and UCLUST. The alpha diversity indices and rarefaction curves were done by in-house code.

## Isolation of lactic acid bacteria from honey

One hundred microliters of each 50% (w/v) honey solution were spread on De Man-Rogosa-Sharp broth (MRS; Oxoid, Basingstoke, UK) (1.5% (w/v) agar) plates and incubated at 30 °C for 48–72 h under microaerophilic conditions. All white, small, round colonies were randomly picked and streaked out on a new MRS plate for pure culture isolation and incubated as described above. All the isolates were stored in 15% (v/v) glycerol at −80 °C for long-term storage.

## Identification of isolates using MALDI-TOF

All isolates were identified using the MALDI-TOF Biotyper (Bruker Daltonics, Bremen, Germany). *L. kunkeei* DSM 12361 or *L. kunkeei* ATCC 700308 was used as a reference strain. The samples were analyzed automatically using Biotyper Compass Explorer software (version 4.1.100) (Bruker, Bremen, Germany). The probability of identification was expressed by a score on a scale from 0 to 3.0. A result above 2.0 denoted secure genus identification and probable species identification. Isolates were selected based on the high probability of identification for further experiments.

## Antimicrobial spectrum

Antimicrobial activity was assessed using an agar well-diffusion method with slight modifications (*Ewnetu, Lemma & Birhane, 2013*). Test organisms belonging to species *Klebsiella spp.*, *Escherichia coli*, *Enterococcus faecalis*, *Pseudomonas aeruginosa,* and *Staphylococcus aureus* were isolated from guts of Deformed wing virus and/or Sacbrood virus-infected *A. cerana* honeybees and identified using MALDI-TOF and 16S rRNA methods by the Laboratory of Molecular Microbiology, IBT, VAST. The turbidity of pathogenic bacterial suspensions, adjusted to match the standard McFarland 0.5 (approximately $10^8$ colony forming units, CFU/mL), was spread onto the plate (*Ewnetu, Lemma & Birhane, 2013*). A 7-mm diameter well was punched aseptically onto the Mueller–Hinton agar (Oxoid, Basingstoke, UK) using the reverse end of a sterile 1-mL pipette tip. A total of 100 μL of test agent (bacterial inoculum, $10^9$ CFU/mL) was seeded into each well. Two negative controls were employed in the assay. The first control involved the MRS medium to account for any inhibition arising from the components of the media. The second control utilized MRS at pH 4 to specifically address inhibition caused by the low pH in the inoculum resulting from the growth of *L. kunkeei*. After incubation at 37 C for 16–24 h, the diameter of the clear zone was measured. All assays were carried out in triplicate.

## 16S rRNA gene sequencing and phylogenetic tree analysis of isolated *L. kunkeei*

DNA extraction of the isolated strains was performed using GeneJET Genomic DNA Purification Kit (Thermo Fisher Scientific, Waltham, MA, USA) according to the manufacturer's instructions. 100 ng of extracted DNA sample was used as a template for the 16S rRNA gene amplification using the primer pair 27F-AGAGTTTGATCMTGGCTCAG and 1492R-GGTTACCTTGTTACGACTT. The PCR reaction was performed in a total volume of 25 μl. The thermal profile for the PCR was 94 °C for 5 min and 35 cycles of 94 ° C for 1 min, followed by 56 °C for 45s and 72 °C for 1 min, and a final cycle at 72 °C for 10 min. The reactions were performed in a C1000 Thermal Cycler (Bio-Rad, USA). The PCR products were electrophoresed on a 1.5% (w/v) agarose gel, stained with ethidium bromide, and visualized under UV light. The PCR products were purified by GeneJET PCR Purification Kit (Thermo Fisher Scientific, Waltham, MA, USA) according to the manufacturer's instructions and directly used for Sanger sequencing by the capillary sequencing system, ABI PRISM 3100 Genetic Analyzer (Applied Biosystems, Waltham, MA, USA).

Sequences were assembled and aligned with Bioedit version 7.0.5.3 (*Hall, 1999*). The nucleotide Basic Local Alignment Search Tool (BLASTn; https://blast.ncbi.nlm.nih.gov/) was used to search the similarity with other sequences deposited in GenBank. The Neighbor-joining phylogenetic tree of the *16S rRNA* sequences of the strains in this study with those obtained from the database was constructed using MEGA X (*Kumar et al., 2018*; *Saitou & Nei, 1987*; *Tamura, Nei & Kumar, 2004*) for evolutionary analysis. The tree was replicated in 1,000 replicates in which the association with taxa clustered together in the bootstrap test.

### Determination of carbon assimilation profiles

Carbohydrate fermentation patterns (APICH50; Biomérieux, Marcy-l'Étoile, France) were finally carried out for *Lactobacillus* to investigate the carbohydrate metabolism of interested isolates according to the manufacturer's instructions. In this assay, API 50 CHL Medium is a ready-to-use medium which allows the fermentation of the 49 carbohydrates on the API 50 CH strip to be studied. Briefly, a suspension is made in the API 50 CHL Medium with each *L. kunkeei* colony to be tested and each tube of the strip is then inoculated with the suspension. During incubation, the carbohydrates are fermented to acids which produce a decrease in the pH, detected by the change in color of the indicator. The results make up the biochemical profile of the strain.

## RESULTS

### Summary of sequencing

Illumina MiSeq paired-end sequencing generated 4,971 valid reads with an average length of 404 bp. Among these sequences, 1,811 (36.4%) were identified at species levels. A total of 134 bacterial operational taxonomic units (OTUs) were identified at the 97% sequence similarity cut-off with Good's coverage of 98.7% and classified into 105 species.

### Alpha diversity indices

The alpha diversity indices of the bacterial community in the honey sample include species richness (ACE = 339, Chao = 205.7, Jackknife = 206.8) and species evenness (Shannon = 2.19, Simpson = 0.242), which are measures of species diversity based on the number and pattern of OTUs observed in the sample, and Phylogenetic diversity (PD = 328) measures the biodiversity incorporating phylogenetic difference between species by calculating the sum of the lengths of all those branches. The result showed that the data sufficiently covered the bacterial components in the sample.

### Diversity of bacteriome in *A. cerana* honey

The EzBioCloud database was used for the taxonomic assignment of 16S rRNA data. Our results showed that the bacterial species in the honey samples collected from 10 different healthy *A. cerana* colonies mainly belonged to two phyla, including *Firmicutes* (50%) and *Proteobacteria* (49%), with two major classes, *Clostridia* (48.2%) and *Gammaproteobacteria* (47.7%), respectively. Some other bacterial phyla were less abundant in the honey sample with a total of ≤ 1%, including *Actinobacteria*, *Acidobacteria*, *Bacteroidetes*, *Planctomycetes*, *Chloroflexi*, *Thermus*, *Nitrospirae*, *Verrucomicrobia*.

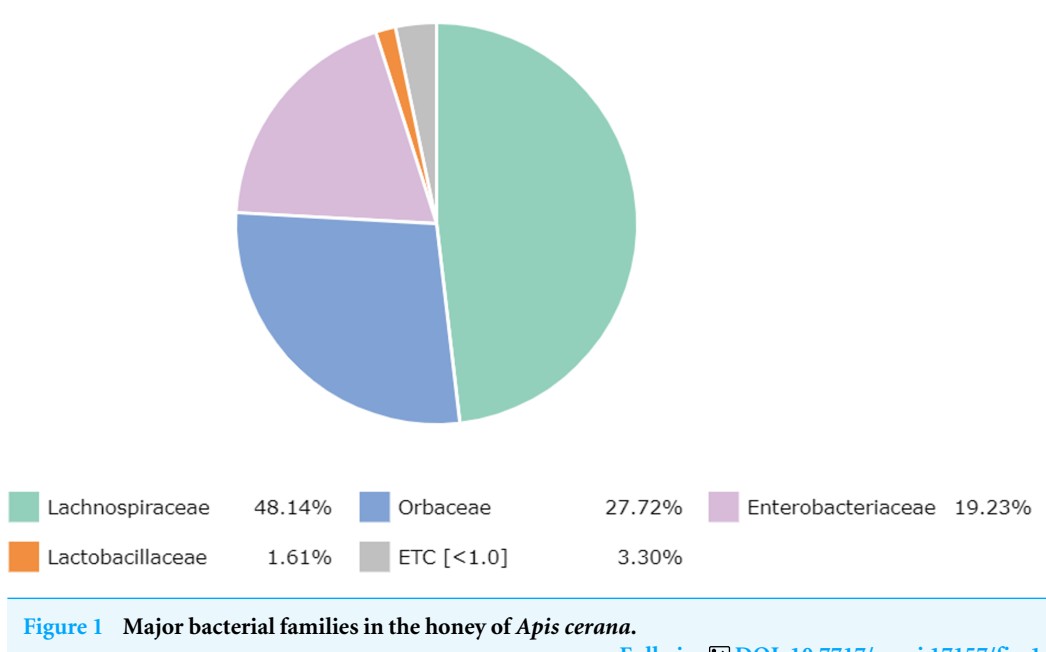

| | | | | | | |
|---|---|---|---|---|---|---|
| ■ Lachnospiraceae | 48.14% | ■ Orbaceae | 27.72% | ■ Enterobacteriaceae | 19.23% | |
| ■ Lactobacillaceae | 1.61% | ■ ETC [<1.0] | 3.30% | | | |

**Figure 1  Major bacterial families in the honey of *Apis cerana*.**

At the family level, there were 65 families detected, with the most abundant family, namely *Lachnospiraceae* (48.14%) (*Clostridia, Firmicutes*), followed by *Orbaceae* (27.72%) (*Orbales, Gammaproteobacteria*), *Enterobacteriaceae* (19.23%) (*Enterobacterales, Gammaproteobacteria*), *Lactobacillaceae* (1.61%) (*Lactobacillales, Firmicutes*) and other bacteria (3.30%) (Fig. 1).

At the genus level, 67 genera were identified in the honey sample, with the three predominant genera accounting for more than 85%, including *Lachnospiraceae_uc* (48.14%) (unclassified species), *Gilliamella* (26.80%), and *Enterobacter* (10.16%); followed by *Enterobacteriaceae_g* (5.77%) (uncultured genus), *Lactobacillus* (1.61%), *Enterobacteriaceae_uc* (1.61%), *Klebsiella* (1.17%), ETC (<1.0%) (4.67%), respectively (Fig. 2). Unfortunately, the most abundant bacterial genus, *Lachnospiraceae_uc,* was all unclassified species. Among 105 species of bacteria identified in the honey sample, *Gilliamella spp* were detected with a total of 26.71%, including a core bacterial group in honeybee gut, *Gilliamella apicola* (13.72%), *Gilliamella JFON_s* (7.54%) (uncultured species) and *Gilliamella_uc* (5.51%) followed by *Enterobacter_uc* (7.52%), *Enterobacteriaceae group* (5.77%), *Lactobacillus spp.* (1.6%) (Fig. 3).

Lactic acid bacteria were detected in the honey sample with a small number (1.63%), which mainly consisted of bacteria belonging to three genera, including *Lactobacillus* (1.6%), *Fructobacillus* (0.08%) and *Weissella* (0.02%). Among identified LAB, the *Lactobacillus kunkeei* group in the *Lactobacillus* genus was the most abundant (1.43%), followed by *Fructobacillus durionis* (0.06%), *Lactobacillus ozensis* (0.06%), *Lactobacillus kimbladii* (0.02%), *Lactobacillus murinus group* (0.02%), *etc.* (Fig. 4).

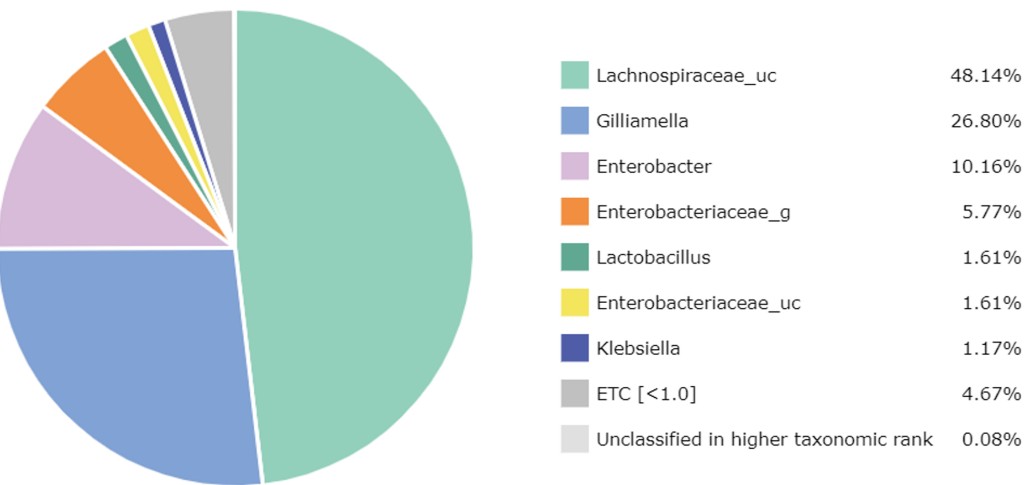

**Figure 2** Bacterial composition at a genus-level in the honey of *Apis cerana* ("uc" indicates unclassified, "g" indicates uncultured genus).

## Isolation and identification of lactic acid bacteria

A total of 56 strains were isolated from the ten honey samples using MRS medium. All isolates were further analyzed by Matrix-assisted laser desorption-ionization time of flight mass spectrometry (MALDI-TOF). The results of the MALDI-TOF analysis revealed seven isolates preliminarily identified as *L. kunkeei* with the highest score of probability of identification, while the remaining isolates belonged to *Enterobacter* spp. *and Klebsiella* spp.

## Antimicrobial spectrum

The agar-diffusion assay was used to "quick screen" the antimicrobial spectra of the seven *L. kunkeei* strains isolated from honey. It was observed that six out of seven *L. kunkeei* strains exhibited antimicrobial activity against all bacterial strains tested, whereas strain *L. kunkeei* LK-VN6 failed to inhibit almost all tested strains except *K. oxytoca* (Table 1, Fig. 5). A negative control (MRS medium) and MRS medium with pH adjusted to 4 had no effect on all pathogenic bacterial strains.

## Molecular identification of selected isolates

Six isolated *L. kunkeei* strains (LK-VN01, LK-VN02, LK-VN03, LK-VN04, LK-VN05, LK-VN07), which have broad antimicrobial spectra, were further identified by 16S rRNA gene sequencing and phylogenic analysis. The results indicated that all the strains collected in this study showed identical 16S rRNA sequences to each other and with 100% identity to *L. kunkeei* (accession number CP128865.1 and CP084249) indicating that they belong to *L. kunkeei* (new nomenclature: *Apilactobacillus kunkeei*). All these sequences have been deposited in the GenBank with accession numbers OR354365, OR365148, OR365149, OR365150, OR365151, OR365152. The 16S rRNA gene sequence of each *L. kunkeei* isolate and those of related species were used to construct the phylogenetic tree. The NJ tree analysis clearly clustered these six strains in one group together with other *L. kunkeei* isolated from
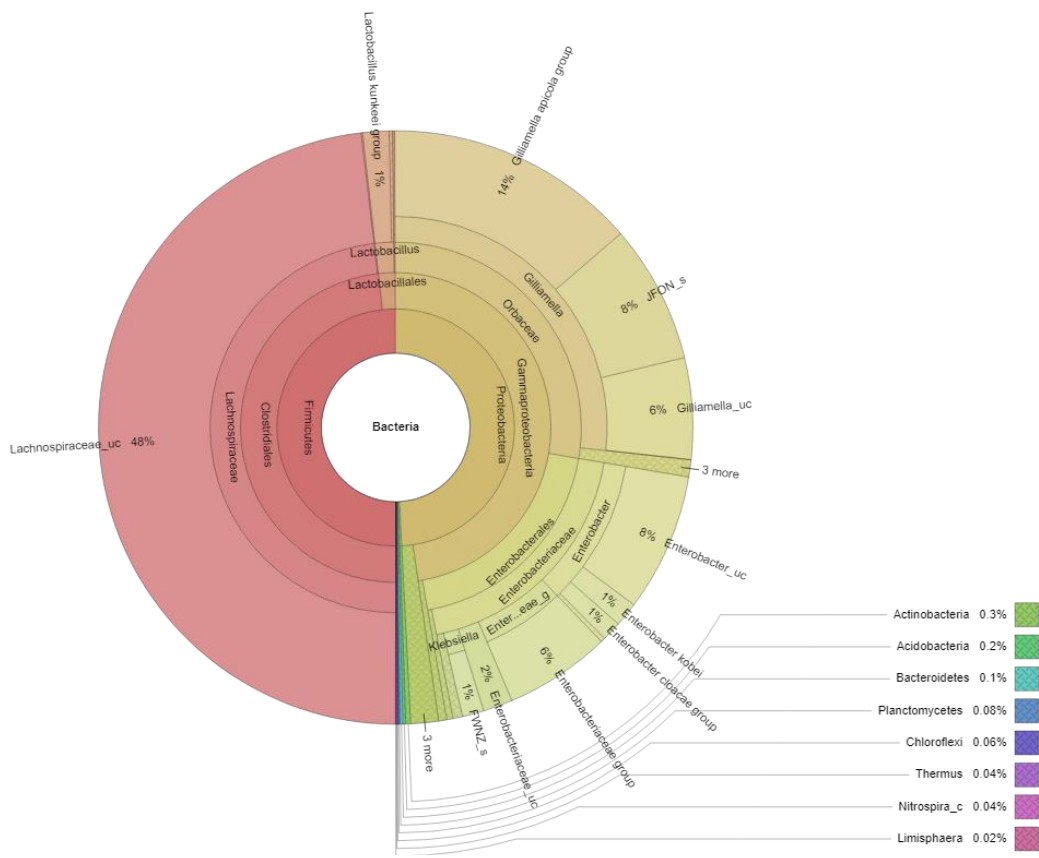

**Figure 3   Krona plot of bacterial community in the honey of *Apis cerana* with a cut-off of 1%.** Notes: "uc" indicates unclassified species, "g" indicates uncultured genus, "c" indicates uncultured clade, "JFON_s" and "FWNZ_s" indicate uncultured species of genera Gilliamella and Klebsiella, respectively.

honey (accession number AB559820.1), beebread (accession number CP128865), and honeybee guts (*A. melifera*) (Fig. 6).

All the strains (VN-LK1, VN-LK2, VN-LK3, VN-LK4, VN-LK5, VN-LK7) were further characterized by the API 50 CH test for carbon source assimilation. The result showed that they fermented 6 of the 49 carbohydrates tested, including glucose, fructose, mannitol, trehalose, sucrose, and gluconate (Table 2).

# DISCUSSION

Honey contains very low water and high sugar content, making it difficult for most microorganisms to grow (*Silva et al., 2017*). However, there are still microorganisms that can survive in honey, including bacteria (*Kačániová et al., 2009*; *Saha, Ahammad & Barmon, 2018*; *Silva et al., 2017*). Recently, by using the culture-dependent method, *Devi et al. (2021)* reported that *A. cerana* honey (Himachal Pradesh, India) had an average bacterial load of 3.74 and 3.99 logs, and it consisted of sufficiently more bacterial diversity than those in *A. mellifera* honey with 57.14% of the isolates being Gram-negative bacteria while the other isolates (42.86%) belonged to Gram-positive bacteria through morphological
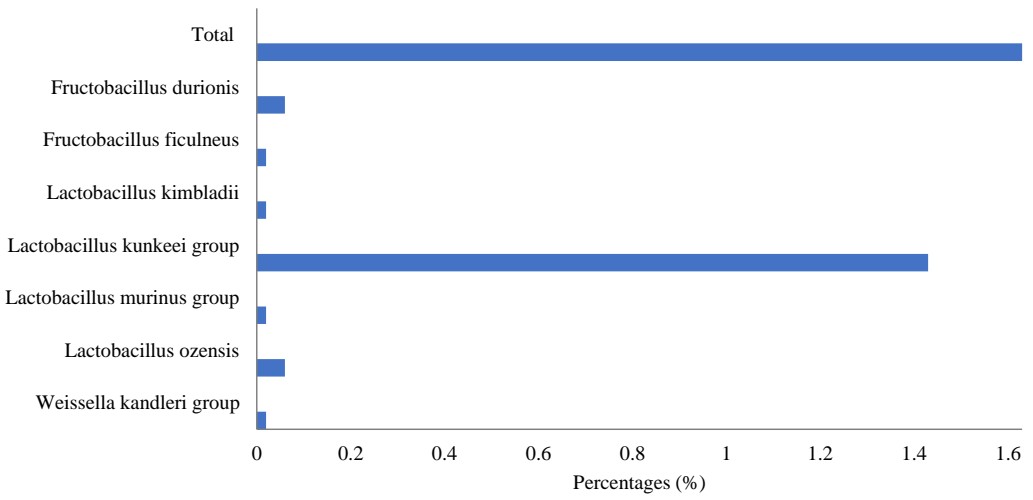

**Figure 4** Lactic acid bacteria in *Apis cerana* honey.

**Table 1** Antimicrobial spectrum of the isolated *L. kunkeei* strains on pathogenic bacteria isolated from guts of illness *A. cerana* honeybee in Vietnam, including *E. coli*, *Klebsiella spp.*, *E. faecalis*, *P. aeruginosa* and *S. aureus*.

| Test strains | L. kunkeei strains | | | | | | |
|---|---|---|---|---|---|---|---|
| | LK-VN01 | LK-VN02 | LK-VN03 | LK-VN04 | LK-VN05 | LK-VN06 | LK-VN07 |
| *E. coli* AC1 | + | + | + | + | + | − | + |
| *E. coli* AC2 | + | + | + | + | + | − | + |
| *K. pneumonia* | + | + | + | + | + | − | + |
| *K. varicola* | ++ | ++ | ++ | ++ | ++ | − | ++ |
| *K. oxytoca* | +++ | +++ | +++ | +++ | +++ | + | +++ |
| *P. aeruginosa* | + | + | + | + | + | − | + |
| *E. facialis* | +++ | +++ | +++ | ++ | +++ | − | +++ |
| *S. aureus* | ++ | ++ | ++ | ++ | ++ | − | ++ |

**Notes.**
*Clear zone around well, +: 1–3 mm, ++: 3–5 mm, +++: >5 mm, -: no inhibition zone was detected.

characterizations (*Devi et al., 2021*). In this study, the bacterial component in the *A. cerana* honey was first evaluated using the NGS technique.

Our results showed that bacterial composition in the *A. cerana* honey sample collected in Hanoi, Vietnam was dominated by only two phyla of *Firmicutes* (50%) and *Proteobacteria* (49%). In contrast, our earlier study on the gut microbiome of adult *A. cerana* honeybees in Hanoi, Vietnam, demonstrated a more diverse profile encompassing four phyla: *Proteobacteria* (70.7%), *Actinobacteria* (10.7%), *Firmicutes* (10.3%), and *Bacteroidetes* (8.3%) (*Duong et al., 2020*). The digestive tract of honeybees is considered one of the most important primary sources of honey microorganisms (*Lee, Churey & Worobo, 2008*; *Silva et al., 2017*). Given this significance, the reduced microbial diversity observed in the honey sample could be attributed to the stringent conditions within honey, potentially inhibiting

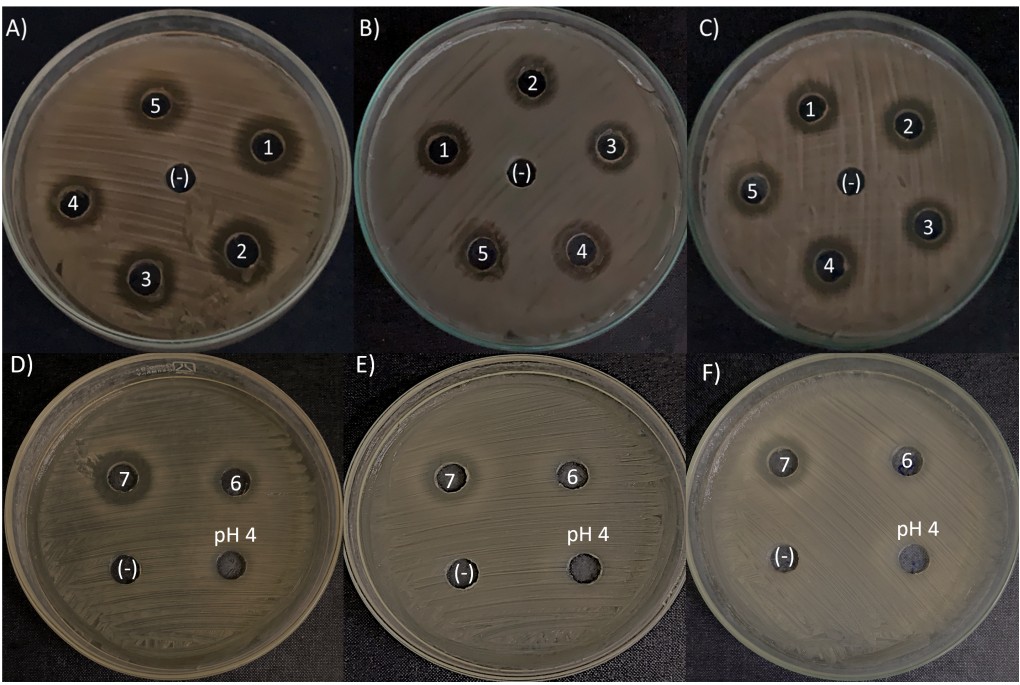

**Figure 5** **Antimicrobial activity exhibited by *L. kunkeei* strains (LK_VN01-VN07) on *K. oxytoca* (A and D), *K. varicola* (B and E), and *S. aureus* (C and F).** MRS medium was used as a negative control (-). MRS at pH 4 to specifically address inhibition caused by the low pH in the inoculum resulting from the growth of *L. kunkeei*.

the viability of various bacteria. This speculation was also supported by the results from previous study (*Silva et al., 2017*).

The abundance of phyla *Firmicutes* and *Proteobacteria* in honey may imply their ability to survival under several environmental pressures (*Ganeshprasad et al., 2022*; *Zhang et al., 2021*), such as an inhospitable conditions in honey. Though most phyla *Firmicutes* and *Alpha-* and *Beta-proteobacteria* utilize sugars and are facultatively anaerobic and acidic-tolerant microbes (*Ganeshprasad et al., 2022*), *Gamma-proteobacteria* include both aerobic and anaerobic bacteria and metabolisms vary among genera (*Zhang et al., 2021*). These bacteria are common in soil, water, and plants, as well as in humans and animals (*Ganeshprasad et al., 2022*; *Zhang et al., 2021*), and some of them, such as *Gamma-proteobacteria*, can be a potential marker for soil pollution in the gut microbiota of the soil invertebrates (*Zhang et al., 2021*).

The honey bacteria in concern are primarily spore-forming bacteria (*Snowdon & Cliver, 1996*). In this study, *Lachnospiraceae* was the most predominant family in the *A. cerana* honey sample (48.14%). They are an abundant family of obligate anaerobic bacteria in the *Clostridiales* order, *Firmicutes* phylum (*Oliphant et al., 2021*; *Vascellari et al., 2020*). This family is considered as core taxa of healthy human gut microbiota, with functions in dietary carbohydrate metabolism with the possibility to ingest polysaccharides, such as starch, inulin, and arabinoxylan; the microbiota's colonization resistance against

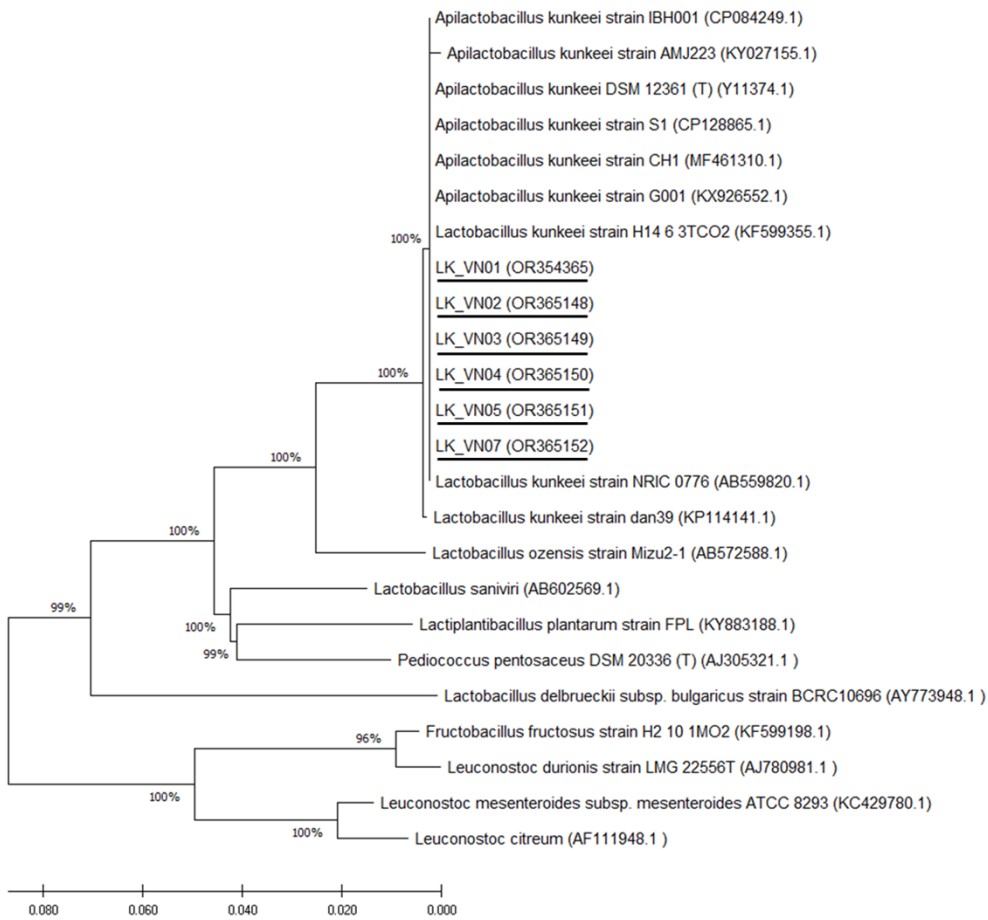

**Figure 6** Phylogenetic tree constructed using the NJ method based on the 16S rRNA gene sequences, indicating the relationships of *L. kunkeei* isolated from *Apis cerana* honey (LK_VN01-LK_VN05 and LK_VN07) to other LAB. The GenBank accession numbers for all the strains are shown in parentheses. Bootstrap values (based on 100 replicates) greater than 70% are indicated at the branching points. The scale bar presents the number of nucleotide substitutions per site. The strains isolated in this study are underlined.

intestinal pathogens through transforming primary to secondary bile acids and production of beneficial metabolites (short-chain fatty acids) and also the host's metabolism and immunity including even epithelium and mucosal immune system (*Sorbara et al., 2020*; *Vacca et al., 2020*; *Vascellari et al., 2020*). The relative abundance of members of the *Lachnospiraceae* family is also associated with several intra and extra-intestinal diseases (*Sorbara et al., 2020*). For example, in the case of Parkinson's disease, the decline of this family exerted significant changes in metabolite production in Parkinson's disease patients (*Vascellari et al., 2020*). Recently, *Lachnospiraceae* members were considered in therapeutic interventions to restore or stimulate microbiota functions (*Sorbara et al., 2020*). Indeed, 4-member consortia of commensal bacteria containing *Lachnospiraceae* can inhibit enteric colonization by vancomycin-resistant *Enterococci* (VRE), *C. difficile*, and *L. monocytogenes*

**Table 2  Biochemical Profile of the API 50 CH Kit of the six representative *L. kunkeei*.**

| $N_0$ | Biochemical tests | Result | $N_0$ | Biochemical tests | Result |
|---|---|---|---|---|---|
| 1 | Glycerol | – | 26 | Salicin | – |
| 2 | Erythritol | – | 27 | Cellobiose | – |
| 3 | D-Arabinose | – | 28 | Maltose | – |
| 4 | L-Arabinose | – | 29 | Lactose | – |
| 5 | Ribose | – | 30 | Melibiose | – |
| 6 | D-Xylose | – | 31 | Sucrose | + |
| 7 | L-Xylose | – | 32 | Trehalose | + |
| 8 | Adonithol | – | 33 | Inulin | – |
| 9 | Methyl xyloside | – | 34 | Melizitose | – |
| 10 | Galactose | – | 35 | D-raffinose | – |
| 11 | D-Glucose | + | 36 | Starch | – |
| 12 | D-Fructose | + | 37 | Glycogen | – |
| 13 | D-mannose | – | 38 | Xylitol | – |
| 14 | Sorbose | – | 39 | Gentibiose | – |
| 15 | Rhamnose | – | 40 | Turanose | – |
| 16 | Dulcitol | – | 41 | Lyxose | – |
| 17 | Inositol | – | 42 | Tagatose | – |
| 18 | Mannitol | + | 43 | D-fucose | – |
| 19 | Sorbitol | – | 44 | L-fucose | – |
| 20 | Methyl-D-mannoside | – | 45 | D-Arabitol | – |
| 21 | Methyl-D-glucoside | – | 46 | L-Arabitol | – |
| 22 | N-acetyl-glucosamine | – | 47 | Gluconate | + |
| 23 | Amygdalin | – | 48 | 2, Keto-gluconate | – |
| 24 | Arbutin | – | 49 | 5, keto-gluconate | – |
| 25 | Esculine | – | | | |

in mice; also, a 12-member consortium can promote resistance against *S. enterica* serovar *Typhimurium (Sorbara et al., 2020)*. The previous study reported that *Lachnospiraceae* were more abundant in *A. mellfera* guts when compared to those in *A.cerana* guts (not detected) (*Ahn et al., 2012*). In our recent studies, *Lachnospiraceae* was about 8% in *A. cerana* pupae but not adults (*Duong et al., 2020*; *Lanh et al., 2022*). It is possible that conditions such as the absence of free oxygen, optimum temperature, *etc.* in *A. cerana* capped brood, and *A. cerana* honey may favor the colonization of these bacteria.

For honey production, honeybees digest nectar with the help of enzymes, and during this process, they also integrate some symbiont microbes associated with the digestive tract that can contribute a positive impact on host health through nourishment activities, including fermentation (*Lee, Churey & Worobo, 2008*). In this study, *G. apicola*, a core bacteria in the digestive tract of adult honeybees (*A. cerana* and *A. melifera*) (*Lanh et al., 2022*; *Silva et al., 2017*), was also found in *A. cerana* honey. It has been reported that *Gilliamella* is a seasonal and nutritional independent bacteria, and they vary due to the sources of nectar and the presence of other bacteria in the gut microbiota of the honeybee (*Silva et al., 2017*).

Lactic acid bacteria have been successfully used as probiotics important in human and farm animal health since they are important components in the gut microbiota of their hosts and with a reported benefit on the gut epithelium system (*Silva et al., 2017*). While the presence of *Lactobacillaceae* in honey was observed at a low abundance (1.61%), it is imperative to recognize that the potential health implications associated with the consumption of *Lactobacillaceae*, even in minute quantities, are diverse. *Lactobacillus* species are widely acknowledged for their probiotic attributes, imparting various health advantages when adequately present in the gastrointestinal tract. These benefits encompass enhanced digestion, immune system modulation, and the prevention of specific gastrointestinal disorders. In this study, lactic acid bacteria were detected, including genera *Lactobacillus*, *Weissella*, and *Fructobacillus,* the most frequent species found in fruits and flowers and in the honey crop of honeybees (*A. mellifera* and *A. cerana*) (*Duong et al., 2020*; *Ruiz Rodríguez et al., 2019*). Due to their fermentation activities, these LAB may play a role in transforming nectar to honey and pollen to beebread (*Silva et al., 2017*). In addition, because honey contains a high concentration of fructose (about 38.5% (*Ewnetu, Lemma & Birhane, 2013*; *Israili, 2014*), some viable bacteria have the capacity to metabolize fructose more easily (*Silva et al., 2017*). In this study, *Lactobacillus kunkeei*, a well-known fructophilic lactic acid bacteria species for their ability to utilize fructose, was detected in a small number, and they were the most abundant LAB in the honey sample.

It has been reported that *L. kunkeei* is a major species in the honey of *Apis* and *Bombus* (*Moran, 2015*) and was the most predominant LAB found in the gut microbiota of *A. melifera* during foraging seasons (*Rangberg et al., 2015*). They were frequently found in honeybees' digestive tracts and hives (*Silva et al., 2017*). In the gut, these bacteria synthesize bacteriocins that inhibit the growth of other microorganisms and improve the host's immune system (*Silva et al., 2017*). In addition, the colonization of *L. kunkeei* inside the hive could eliminate the contamination of fungi that can spoil the nectar, maintaining the organoleptic properties and nutrients of honey (*Arredondo et al., 2018*).

A previous study showed that the genus *Bacillus* was the predominant inhabitant of *A. cerana* honey from Himachal Pradesh, India (*Devi et al., 2021*). Moreover, it has been reported that *Parasaccharibacter apium* was one of the most abundant bacterial species in the honey of *Apis* and *Bombus* species (*Moran, 2015*). In addition, in *Saha, Ahammad & Barmon (2018)*, the predominant type of bacteria commonly found in raw and commercial honey produced by Bangladesh *A. mellifera* were *Streptococci*, *Staphylococci*, *Micrococci*, *Bacilli*, *Lactobacilli*, *Escherichia coli*, *Klebsiella*, *Pneumonia* and gram-negative *Micrococcus luteus*. More recently, the analysis of the 16S rRNA gene in the honey of *Heterotrigona itama* from Malaysia revealed that most of the isolated species belonged to *Bacillus sp.*, *Pantoea sp.,* and *Streptomyces sp.* (*Ngalimat et al., 2019*). However, these bacterial species were not found or presented in small percentage (less than 0.1%) in our study. It could be speculated that factors such as the species of bees, the floral source used by the honeybee, the geographical features of Vietnam, and even honey production processing and storage conditions may contribute to the bacterial composition of honey from Vietnam.

Honey has been known for its antimicrobial properties against food spoilage and pathogenic microorganisms (*Lee, Churey & Worobo, 2008*; *Schell et al., 2022*). A previous

study reported that both stingless bee's honey and *A. mellifera* white honey in Ethiopia exhibited superior antibacterial activity against *S. aureus* (ATCC 25923), *E. coli* (ATCC 25922), and resistant clinical isolates, such as Methicillin-resistant S. *aureus* (MRSA), *E. coli* (R) and *K. pneumoniae* (R) and these honeys were the most common antibiotics used to kill bacteria, making them a novel source of chemotherapeutic agents to combat drug-resistant bacteria in future (*Ewnetu, Lemma & Birhane, 2013*). Furthermore, it has been suggested that the production of antimicrobial compounds by the bacteria associated with honey may take a competitive advantage over other strains in the honeybee gut, selecting antimicrobial-producing bacteria over non-producing sensitive bacteria (*Lee, Churey & Worobo, 2008*). As *Lee, Churey & Worobo (2008)* reported that 92.5% (2,217 out of 2,398) of all bacterial strains isolated from the honey samples were able to synthesize antagonistic compounds against the tested foodborne pathogens and food spoilage microorganisms (*Lee, Churey & Worobo, 2008*). Therefore, honey-derived bacteria possessing antimicrobial activity can be inspected as a new source of antimicrobial compounds in the food industry and for medicinal purposes (*Lee, Churey & Worobo, 2008*; *Okamoto et al., 2021*). Indeed, it has been suggested that various microorganisms in the honey and the gut of honeybees have antagonistic effects on several honeybees and human pathogens, including *Bacillus* genus, lactic acid bacteria such as *Lactobacillus*, *Enterococcus*, *Bifidobacteria*, and *Acetobacteraceae* (*Schell et al., 2022*; *Silva et al., 2017*); therefore same microorganisms can be explored for different purposes, such as its potential as fermenters or probiotics (*Silva et al., 2017*). In the present study, with the ultimate aim of isolating bacterial strains with potential probiotic applications in honeybees and humans, drawing on the NGS results, our efforts were directed toward isolating lactic acid bacteria with promising characteristics. The isolated strains were initially screened by MALDI-TOF identification. As a result, among these, seven strains were identified as *L. kunkeei*. This bacterium was previously reported to be isolated from honey (*Endo et al., 2012*), the guts of *A. melifera* (*Janashia et al., 2016*; *Pachla et al., 2018*), bee bread (*Janashia et al., 2016*), honeybee hive (*Daisley et al., 2020*), *etc.* Biochemically, sugar fermentation patterns of the *L. kunkeei* isolates from Vietnam *A. cerana* honey were fructose, glucose, mannitol, trehalose, sucrose, and gluconate, which were similar to those recorded for the *L. kunkeei* bacteria (*Endo et al., 2012*; *Pachla et al., 2018*). Sequence analysis of the 16S rRNA gene showed that six isolates (excluding LK_VN06) shared 100% sequence similarity with each other and identical (100%) to *L. kunkeei* DSM 12361 (accession number Y11374.1), *L. kunkeei* strain isolated from Japan honey (accession number AB559820.1) and other seven *L. kunkeei* presented in NJ phylogenetic tree. None of the other related species shared 16S rRNA gene sequence similarities higher than 95% with the six isolates. The six isolates and nine different *kunkeei* strains formed an independent subcluster in the *Lactobacillus* phylogenetic group. Hence, from this study, in addition to molecular identification, the use of MALDI-TOF is an effective and rapid method for identifying *L. kunkeei* species found in honey.

*L. kunkeei* has been shown to have a broad spectrum of inhibition *versus* human and honeybee pathogens (*Berríos et al., 2018*; *Butler et al., 2016*; *Lashani, Davoodabadi & Dallal, 2020*; *Olofsson et al., 2016*). In this study, six out of seven isolated *L. kunkeei* strains showed antimicrobial activity against all bacterial strains tested, including *Klebsiella spp.*, *E. coli*,

*E. faecalis*, *P. aeruginosa* and *S. aureus*, which were pathogenic bacteria derived from the illness *A. cerana* honeybee guts. In contrast, no inhibition of pathogenic bacteria was observed in the controls (fresh MRS medium and MRS medium with pH 4). This data suggested that the isolated strains produce antimicrobial compounds, such as bacteriocins that inhibit pathogenic bacteria. Our results are consistent with a previous study which indicated that *L. kunkeei* strains isolated from *A. melifera* guts showed antimicrobial activity against the major honeybee pathogens (*Lashani, Davoodabadi & Dallal, 2020*), *Paenibacillus larvae*, *E. coli* ATCC 25922 and *K. pneumoniae* ATCC 700603 (*Pachla et al., 2018*). Moreover, *L. kunkeei* strains isolated from honey and honeybee guts have been shown to harbor several other natural probiotic properties, such as resistance to acid and bile (*Lashani, Davoodabadi & Dallal, 2020*), good cell–surface properties (hydrophobicity, auto-aggregation, and biofilm production) (*Berríos et al., 2018*; *Iorizzo et al., 2020*) and good resistance to high sugar concentrations (*Iorizzo et al., 2020*). In fact, as an important member of the LAB group, *L. kunkeei* is a well-known probiotic microorganism (*Ramos et al., 2020*). The supplement *L. kunkeei* only or combined with other LAB can decrease the mortality rate and significantly enhancement the longevity of the honeybees, induce immune stimulation, and increase honey production (*Iorizzo et al., 2022*; *Rangberg et al., 2015*), mitigate antibiotic-associated microbiota dysbiosis and immunodeficiencies in honeybees (*Daisley et al., 2020*). Moreover, recent studies have indicated many proteins produced by these bacteria are able to act as antimicrobial compounds against pathogens causing human wound infections (*Butler et al., 2016*; *Olofsson et al., 2016*; *Schell et al., 2022*). Taken together, the *L. kunkeei* species found in honey can be promising candidates for developing probiotics for honeybee and human uses.

The present study, while providing valuable insights into the microbiota of honey, is not without its limitations. A notable limitation of the present study lies in its reliance on samples collected in only one province and it may not adequately capture the diversity and variability that could exist within different batches or sources of honey of Vietnam. In addition, honey production can be influenced by various factors such as geographical location, floral sources, and beekeeping practices, all of which may contribute to distinct microbial compositions. Therefore, future studies should incorporate a more extensive and diverse set of samples, encompassing different regions and honey types. Previous studies have already identified microbial communities in honey; however, they focused on isolation of bacteria in honey and/or other honeybee species. In this study, the microbial composition in *A. cerana* honey in Hanoi, Vietnam was first investigated by NGS technology combined with isolation of bacteria.

## CONCLUSION

Our findings revealed the diversity of the bacterial community in *A. cerana* honey collected from Hanoi, Vietnam, including the presence of various beneficial bacteria with implications for humans, insects' well-being, and other animals. These identified bacteria were *Lachnospiraceae* spp., *Gilliamella* spp., *L. kunkeei*, *F. durionis*, *L. ozensis*, *L. kimbladii*, and *L. murinus*. Notably, the *L. kunkeei* strains isolated from honey produced by

*A. cerana* in Vietnam exhibited a broad antimicrobial spectrum against various pathogenic bacteria in honeybees. This suggests the potential of these bacteria as promising probiotic candidates due to their natural origin and ability to inhibit the colonization of honeybee pathogens. The *in vivo* studies are needed to determine the effect that honey microbes may have on honeybee health.

## Abbreviations

| | |
|---|---|
| **DNA** | deoxyribonucleic acid |
| **PCR** | polymerase chain reaction |
| **RT-PCR** | reverse transcriptase polymerase chain reaction |
| **rRNA** | ribosomal ribonucleic acid |

## ACKNOWLEDGEMENTS

The authors would like to thank Dr. Pham Hong Thai from the Research Center for Tropical Bees and Beekeeping, Vietnam National University of Agriculture, for providing honey samples.

### Funding

This study was supported by the Ministry of Science and Technology of Vietnam, grant No. DTDL.CN-44/21. The authors declare that no funds, grants, or other support were received during the preparation. The funders had no role in study design, data collection and analysis, decision to publish, or preparation of the manuscript.

### Grant Disclosures

The following grant information was disclosed by the authors:
Ministry of Science and Technology of Vietnam:  DTDL.CN-44/21.

### Competing Interests

The authors declare there are no competing interests.

### Author Contributions

- Pham T. Lanh performed the experiments, analyzed the data, prepared figures and/or tables, authored or reviewed drafts of the article, and approved the final draft.
- Bui T.T. Duong performed the experiments, prepared figures and/or tables, and approved the final draft.
- Ha T. Thu performed the experiments, prepared figures and/or tables, and approved the final draft.
- Nguyen T. Hoa performed the experiments, prepared figures and/or tables, and approved the final draft.
- Dong Van Quyen conceived and designed the experiments, analyzed the data, authored or reviewed drafts of the article, and approved the final draft.

## DNA Deposition

The following information was supplied regarding the deposition of DNA sequences:

16S rRNA of six L. kunkeei strains are available at the NCBI database: OR354365, OR365148, OR365149, OR365150, OR365151, OR365152.

## Data Availability

The sequence and sample data are available at the NCBI database: PRJNA943093.

The raw data is available in the Supplementary File.

## Supplemental Information

Supplemental information for this article can be found online at http://dx.doi.org/10.7717/peerj.17157#supplemental-information.

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
