# Peer review of "Comprehensive analysis of the microbiome in Apis cerana honey highlights honey as a potential source for the isolation of beneficial bacterial strains"

_PeerJ, doi:10.7717/peerj.17157_

## Round 0.1 · original submission · Major Revisions

Please follow the comments of the reviewers to revise your manuscript. A point-by-point response letter is needed when resubmitting your manuscript.

Reviewer 1 ·

Basic reporting

Several edits are required.
Title: "Comparitive" this study is not a comparitive study

Page 4. Results section : "LAB species" please elaborate meaning
L82-87 runon-on sentense. Needs a break at L84 after "skincare ingredient"
L88-91 tense confusion- paragraph starts in past tense "study aimed" and continues in future "attnetion will". Parhaps present tense would work best for this paragpraph.

L93 "The obtained results will provide" - this is a strong statement that does not have concreate evidence, soften the statement with a "may" or "could". e.g. the results in this study could be used to develop...
L108 should read "All pairs of primers are in Table S1."
L298 - typo "psychological"
L335 - wrong word use "adaquate"

Experimental design

Several clarification regarding the method sections are required:

Were the honey samples collected all at the same time? were they tested immediately after collection? if not, how were they stored?

after PCR analysis, samples were stored in room temperature, how long were they stored for before micro analysis? Could the bacterial composition change in that time? include in discussion
L194 - API 50 CH test for carbon source assimilation needs more details in methods
Results
L258 - NJ tree analysis - please include how this analysis was performed in the methods section

Validity of the findings

no comment

Reviewer 2 ·

Basic reporting

This article meets the requirements for basic reporting. There are minor issues with English language which do not affect the reporting, but which are outlined in my comments below.

Experimental design

1. The knowledge gap for performing the 16S rRNA gene sequencing study is not adequately outlined in the introduction. Previous studies have identified microbial communities in honey in Apis cerana, as cited in the introduction and discussion, however the authors present this as the first study to do this in Lines 88-90. Perhaps they should reframe it to identify microbial communities in Vietnam specifically.

Methods related to the antimicrobial spectrum and carbohydrate fermentation patterns need more details, as follows:

2. L171- “100 ul of test agent was seeded into each well”—what is the test agent made up of? Is it bacteria in media? What concentration? The authors should also note here that a negative well was made. What was seeded into the negative control? If it was just an empty well, how do the authors know that inhibition didn’t occur due to other substances in the test agent (e.g. media)?

3. L194 – Please provide information on how the carbohydrate fermentation patterns were tested.

Validity of the findings

The manuscript meets the validity of findings.

Additional comments

The manuscript “Comparative analysis of the microbiome in Apis cerana honey highlights honey as a potential source for the isolation of beneficial bacterial strains” by Lanh et al uses 16S rRNA gene sequencing to identify gut microbes present in Apis cerana honey from bees from Hanoi, Vietnam. The authors found that the microbiota of this honey sample consisted of species within the phyla of Firmicutes and Proteobacteria. Focusing their analysis on lactic acid bacteria, they found that the most highly abundant LAB was Apilactobacillus kunkeei. Subsequent isolation of 7 strains of this bacteria followed by antimicrobial activity tests showed that 6 out of the 7 strains of A. kunkeei inhibited growth of common and coinciding honey bee bacterial pathogens.

Overall I enjoyed reading this study and found novelty in their study isolating A. kunkeei strains and testing their ability to inhibit growth against pathogens that were isolated from their own colonies. However, the following points should be addressed:

Comments to Introduction:
1. The knowledge gap for performing the 16S rRNA gene sequencing study is not adequately outlined in the introduction. Previous studies have identified microbial communities in honey in Apis cerana, as cited in the introduction and discussion, however the authors present this as the first study to do this in Lines 88-90. Perhaps they should reframe it to identify microbial communities in Vietnam specifically.
2. L87- end this paragraph noting that understanding antimicrobial activity of the microbes in honey could be better understood. This is the novelty of this study.
3. Here and throughout the manuscript Lactobacillus kunkeei should be changed to the new nomenclature of Apilactobacillus kunkeei.
4. L93-95 – This sentence is too strong. This manuscript does not show that honey microbes can affect honey bee health. The authors should change this to “The obtained results have the potential to provide insight into developing biological products…”

Comments to Methods:
5. L98 – where did the A. cerana colonies originate from? Were they provided fresh frames to build comb and store honey in? Since comb can have an effect on microbiota it is important to note that the wax and honey did originate from the colonies that they took them from.
6. L109 – How long was the honey stored, and how would storage effect the microbiota?
7. L117—were the DNA samples pooled in equal amounts? If not, how much of each?
8. L134-148 – The authors should include in methods how they analyzed the data at the family level, and then broke down this data into genera, etc.
9. L167 – “the turbidity of bacterial suspensions”- are the authors referring to the pathogenic bacteria or the A. kunkeei isolates?
10. L171- “100 ul of test agent was seeded into each well”—what is the test agent made up of? Is it bacteria in media? What concentration? The authors should also note here that a negative well was made. What was seeded into the negative control? If it was just an empty well, how do the authors know that inhibition didn’t occur due to other substances in the test agent (e.g. media)?
11. L174- Should say “Isolated A. kunkeei full 16S rRNA gene amplifying and sequencing” or something similar to distinguish this section from the Next generation sequencing.
12. L194 – Please provide information on how the carbohydrate fermentation patterns were tested.

Comments to Results:
13. L212- “Our results showed that the bacterial species in the honey sample collected from a healthy A. cerana colony…”—this sentence makes it seem that only one sample was taken from one colony, but the methods say 10 samples were taken. The authors should be very clear about how many samples. If it is only 1 sample, the authors should include discussion of this limitation in the discussion section.
14. Figure 5 – only 5 isolates are shown in the figure, but the authors tested 7. The figure should be updated to show all 7.

Comments to Discussion:
15. The Discussion section needs to include discussion of the two main limitations of this study: that it is based on a single sample, and that it is not the first study to identify gut microbiota of honey. These should be stated clearly and the implications of each should be outlined.
16. The Discussion would also benefit from discussion of the fact that Lactobacillaceae were in low abundance (1.61%) in honey. Given that the authors focus on LABs within this group and their potential as probiotics, are these in high enough concentration in honey to really have an effect on human or animal health?
17. L268-285- This section may fit better in the introduction than in the discussion since it focuses on what was already known about honey microbiota prior to this study.
18. L294 “Our data was further supported…”- does this refer to the present data, or the data from your previous study?
19. L296-298- please clarify this sentence. It is unclear of the authors mean that the bacteria have different functions, or whether the environment changes the microbiota present.
20. L335 – here and in other places in the discussion (e.g. L369) the authors refer to “adequate” or “inadequate” amounts of bacteria. How is an adequate amount being determined? It would be better to say the percentage of the relevant taxonomic group in the honey sample rather than and “adequate amount.”
21. L398-399 “These findings underscore that honey provides an inhospitable environment for bacterial growth.”—this sentence is confusing. The authors were able to sequence and isolate bacteria from honey, so this sentence does not seem to be true. The authors should edit this out or change it to be more precise.
22. L447-449 – The authors should note that in vivo studies (in honey bees) are needed to determine effect that honey microbes have on honey bee health.

The following are minor comments to sentence structure and grammar:

23. L20 – “nutrient food” should be “nutritious food”
24. L44 – “produced by bees, an important” should be “produced by bees, and an important”
25. L48 – “Honey for human consumption is delivered mainly” should be “Honey for human consumption is mainly produced by”
26. L69 – “more important” should be “most important”
27. L70 – “food, commonly colonized” should be “food, and they are commonly colonized”
28. L71-76 – This sentence is too long and confusing. It would help to be split into two, after “long-term honey storage” (L73).
29. L91 – “attention will be given” should be “attention was given”
30. L101 – “for the free” should be “to be free”
31. L107 – “PCR as described previous study” should be “PCR as described in a previous study”
32. L108 – “All pairs of primers were in Table S1” should be “All pairs of primers are in Table S1”
33. L120-121 – “including primers of 341F” should be “using primers 341F”
34. L283 – “isolates were Gram-negative” should be “isolates being Gram-negative”
35. L286-290—change this sentence to: “Our results showed that bacterial composition in the honey sample from A. cerana bees from Hanoi, Vietnam was dominated by only two phyla, Firmicutes (50%) and Proteobacteria (49%), while our previous study showed that the gut microbiota in adult A. cerana bees from Hanoi, Vietnam generally consisted of four phyla, including Proteobacteria (70.7%), Actinobacteria (10.7%), Firmicutes (10.3%), and Bacteroidetes (8.3%).
36. L300-302- change this sentence to “Gammaproteobacteria include both aerobic and anaerobic bacteria and metabolisms vary among genera.”
37. L304-305: change to “and some of them, such as Gammaproteobacteria, can be potential markers for soil pollution in the gut microbiota of soil invertebrates.”
38. L338 – “to bring advantages to” should be “important in”
39. L339 – “ingredients” should be “components”
40. L344 – “their function” should be “a role”
41. L349 – “detected with a small number and they were” should be “detected in a small amount and were”
42. L356 – “Besides” should be “In addition”
43. L369 – “This could be speculated” should be “It could be speculated”
44. L378 – change to “and these honeys were the most common antibiotics used to kill bacteria, making them a novel source”
45. L440-442- restructure this sentence to: “Our findings revealed the diversity of the bacterial community in A. cerana honey collected from Hanoi, Vietnam, including the presence of various beneficial bacteria with implications for human, insect, and other animal well-being.”

---

## Round 0.2 · Minor Revisions

Please revise the manuscript by following the reviewer's comments.

Reviewer 1 ·

Basic reporting

Data Availability statements were deleted. Please include them in the manuscript.

Experimental design

no comment

Validity of the findings

no comment

Additional comments

The authors positively addressed the comments.

Reviewer 2 ·

Basic reporting

No comment.

Experimental design

No comment.

Validity of the findings

No comment.

Additional comments

Thank you for making my suggested changes. The manuscript looks great.

---

## Round 0.3 · accepted · Accept

Thanks for addressing all the comments and suggestions. I would like to endorse the publication of the manuscript.